# Genomic and Transcriptomic Underpinnings of Melanoma Genesis, Progression, and Metastasis

**DOI:** 10.3390/cancers14010123

**Published:** 2021-12-28

**Authors:** Olga S. Cherepakhin, Zsolt B. Argenyi, Ata S. Moshiri

**Affiliations:** 1School of Medicine, University of Washington, Seattle, WA 98195, USA; olgacher@uw.edu; 2Department of Laboratory Medicine and Pathology, University of Washington, Seattle, WA 98195, USA; zsolt@uw.edu; 3Division of Dermatology, Department of Medicine, University of Washington, Seattle, WA 98195, USA

**Keywords:** metastasis, melanoma, genomics, pathways, transcriptomics, oncogenesis, MAPK, BRAF

## Abstract

**Simple Summary:**

Melanoma is a skin cancer with a high mortality and a dramatically rising presence worldwide. Recent research has shed new light on the genetic events that promote melanoma progression and confer metastatic potential. This review summarizes the role of molecular pathways, genomic factors, and the tumor microenvironment in the progression from local melanoma to distant disease. Further characterization of these elements is necessary to identify relevant prognostic factors and potential new therapeutic targets.

**Abstract:**

Melanoma is a deadly skin cancer with rapidly increasing incidence worldwide. The discovery of the genetic drivers of melanomagenesis in the last decade has led the World Health Organization to reclassify melanoma subtypes by their molecular pathways rather than traditional clinical and histopathologic features. Despite this significant advance, the genomic and transcriptomic drivers of metastatic progression are less well characterized. This review describes the known molecular pathways of cutaneous and uveal melanoma progression, highlights recently identified pathways and mediators of metastasis, and touches on the influence of the tumor microenvironment on metastatic progression and treatment resistance. While targeted therapies and immune checkpoint blockade have significantly aided in the treatment of advanced disease, acquired drug resistance remains an unfortunately common problem, and there is still a great need to identify potential prognostic markers and novel therapeutic targets to aid in such cases.

## 1. Introduction

The progression from melanomagenesis to metastasis is known to differ between the various subtypes of melanoma, which are defined by their disparate clinical, histopathologic, and genetic features. In 2018, the 4th Edition of the World Health Organization Classification of Skin Tumours identified 10 distinct melanoma pathway subtypes, based largely on the role of pathogenic ultraviolet light (UV) light exposure, as well as the characteristic driver mutations often found in each pathway [1]. This reclassification has revolutionized our understanding of melanoma, at once deemphasizing the traditional clinical and histomorphologic changes that have been historically important while bringing to the surface the molecular and genetic aberrations now recognized to underly them. While efforts to date have largely clarified the important molecular events in melanogenesis, more recent research has shed new light on the molecular evolution of metastasis in melanoma, and in particular uveal melanoma. In this review we will discuss the genomic and transcriptomic pathways of melanoma, important drivers of melanoma metastasis, and the role the tumor microenvironment plays in metastatic progression.

## 2. Genomic and Transcriptomic Pathways of Melanomagenesis and Progression

### 2.1. UV Signature Mutations

UV radiation exposure is an important risk factor for most subtypes of cutaneous melanoma [1,2]. UV signature mutations of cytosine-to-thymine (C-to-T and CC-to-TT) account for about 45% of mutations in cutaneous melanomas while guanine-to-thymine (G-to-T) substitutions account for 10% [3,4,5]. Acral, uveal and mucosal melanomas have the lowest rates of UV signature mutations suggesting UV exposure has little to do with the pathogenesis of these subtypes [6].

### 2.2. MAPK/ERK Pathway

The MAPK/ERK (mitogen activated protein kinase/extracellular signal regulated kinase) pathway is important for cell survival and proliferation and has been heavily implicated in the development and progression of cutaneous melanoma [7]. Mutations that upregulate the pathway, including BRAF (B rapidly accelerated fibrosarcoma kinase), RAS (rat sarcoma virus small GTPase), and NF1 (neurofibromin 1), are found in more than 80% of cutaneous melanomas [8]. Low-CSD melanomas have a moderate mutational burden and are associated with BRAFV600E mutations while high-CSD melanomas have a higher mutational burden and are associated with NRAS, NF1, and BRAFnonV600E mutations [9]. Cutaneous melanomas can be classified into four main genetic subtypes: BRAF mutated, RAS mutated, NF1 mutated, and triple-wildtype [6].

#### 2.2.1. BRAF

Mutations in BRAF result in the constitutive activation of the MAPK/ERK pathway independent of RAS activity [10,11]. The most commonly-altered gene in cutaneous melanoma is BRAF, and mutations and oncogenic fusions involving BRAF are present in 40–60% of cutaneous melanomas, spanning most histologic subtypes including superficial spreading, lentigo maligna, acral lentiginous, desmoplastic, Spitz, and those arising from congenital nevi [1,10,12]. Mutations in BRAF result in the constitutive activation of the MAPK/ERK pathway independent of RAS activity [10,11]. The BRAF V600E substitution is the most common driver mutation in superficial spreading melanoma and it accounts for up to 90% of BRAF mutations in cutaneous melanoma, with V600K accounting for a significant remainder [1,11].

#### 2.2.2. RAS

When mutated, NRAS remains in the perpetually activated state, resulting in the constitutive activation of MAPK/ERK signaling as well as PI3K/AKT signaling [5,10]. NRAS is mutated in 15–25% of cutaneous melanomas and is the most common driver mutation seen in melanoma arising from giant congenital nevi [5,6,7,10]. When mutated, NRAS remains in the perpetually activated state, resulting in the constitutive activation of MAPK/ERK signaling as well as PI3K/AKT signaling [5,10]. Mutations in the less commonly affected oncogenes of HRAS have been identified in Spitz neoplasms (both benign and malignant), while and KRAS has been identified as an important driver in acral lentiginous and mucosal melanomas [1,6]. Generally, BRAF and RAS mutations are thought to be mutually exclusive and are correlated with different clinical and histopathological presentations, though they can rarely be found together in separate clonal subpopulations of a given tumor [12].

#### 2.2.3. NF1

Loss of function in NF1 prevents its function as a GTPase-activating protein (GAP), thus resulting in the upregulation of MAPK/ERK signaling as well as PI3K/AKT signaling [7]. Neurofibromin 1 (NF1) is a tumor suppressor, and loss of function mutations occur in this gene in 10–15% of melanomas and are especially prevalent in high-CSD melanomas (i.e., lentigo maligna) [1,5,6,7,9]. Loss of function in NF1 prevents its function as a GTPase-activating protein (GAP), hence resulting in the upregulation of MAPK/ERK signaling as well as PI3K/AKT signaling [7]. Though NF1 mutations often occur in melanomas without BRAF or NRAS mutations, approximately 4% of the BRAF and NRAS mutated melanomas also harbor NF1 mutations [7,13].

#### 2.2.4. Triple Negative

It is estimated that about a third of cutaneous melanomas are triple-negative and have no BRAF, NRAS, or NF1 mutations [7]. Triple-wildtype melanomas are more likely to be non-CSD melanomas [6,14]. Alterations in other genes in the MAPK/ERK pathway in melanoma include RASA2, PTPN11, MEK1, and MEK2 [1,15,16].

The mutation statuses of melanomas have been shown to influence the development of metastasis, but there are conflicting results between studies as to metastatic potential [12]. Broekaert et al. found that BRAF-mutant melanomas favor nodal metastasis while BRAF-wildtype melanomas are more likely to metastasize to non-nodal sites [17]. These results are in agreement with Mar et al.’s large cohort study, which ascertained that BRAF mutation correlated with a higher probability of lymph node metastasis at the time of diagnosis [18]. Barbour et al., on the other hand, found that primary recurrences of stage III BRAF-mutant melanomas were almost always distant metastases and that confined regional nodal metastases were very infrequent [19]. Consistent with Barbour et al., Chang et al.’s small retrospective study showed that BRAF-mutant melanomas had a higher probability of metastasizing to the liver as compared to BRAF-wildtype melanomas [20].

Conflicting data has also been published regarding the role of both BRAF and NRAS mutations in melanoma metastasis. A more recent prospective large cohort study by Adler et al. found that both NRAS and BRAF mutation statuses in melanomas were associated with CNS and liver metastasis, that NRAS alone correlated with liver metastasis and that BRAF alone was associated with first metastasis to the lymph nodes [21]. Jakob et al., on the other hand, found that BRAF-mutant melanomas were not associated with liver or nodal metastasis but rather with much higher rates of CNS and lung metastasis [22]. Additionally, they found that CNS involvement was also more likely for NRAS-mutant melanomas at time of distant disease diagnosis [22]. Doma et al. found that BRAF mutant allele fractions were significantly enhanced in the lung, adrenal gland, intestine, and kidney metastases, but not in the CNS or liver [23]. Though there is disagreement on the specific correlations between melanoma metastasis and BRAF/NRAS mutation status, BRAF and NRAS mutations are known to be stage-independent risk factors for worse prognosis in the metastatic setting [24]. In considering these data, it is important to note that there can be discrepancies in BRAF and NRAS mutation statuses between metastatic lesions and primary and metastatic lesions of the same tumor, and that biopsy and genetic analysis of both primary and metastatic lesions may be informative in considering prognosis and treatment options [25,26].

Though MAPK/ERK pathway mutations are prevalent in cutaneous melanoma, they are likely insufficient for melanomagenesis [14]. BRAF and NRAS mutations are common in benign nevi, suggesting that additional mutations are required to drive melanoma progression in those tumors [14,27,28].

### 2.3. PI3K/AKT Pathway

The PI3K/AKT pathway is known to promote cell proliferation, metabolism, motility, angiogenesis, and survival, as well as mTOR signaling [7]. Up to 70% and 15% of cutaneous melanomas have mutations affecting the PI3K/AKT and mTOR pathways, respectively [6,29]. PI3K/AKT signaling can be activated by mutations in NRAS and NF1 as well as those in other genes [5,7]. PTEN is a tumor suppressor that dephosphorylates PIP3 to PIP2, preventing the activation of AKT [5,7]. Silencing of PTEN has been found in 10–30% of cutaneous melanomas [5,7]. Mutations in PTEN are often found to occur in the later stages of primary cutaneous melanoma invasion and loss of PTEN expression in clinical stage III melanomas is associated with a shorter time to brain metastasis [14,30]. A study of conjunctival melanoma by Kenawy et al. demonstrated that 10q11.21-26.2 chromosome deletions, which include PTEN, are also associated with melanoma metastasis [31]. Alterations in other important constituents of the PI3K/AKT pathway, namely RAC1, PIK3CA, AKT3, PREX2, and AKT1, are also known to occur in melanoma [1,5,8,32,33]. Phosphorylated AKT was discovered to have a significantly higher expression in CNS metastases [34]. and the activating E17K mutation in AKT1 was recently shown to promote brain metastasis formation and reduce survival in a melanoma mouse model study [29]. Other genes that have been implicated in promoting both PI3K/AKT and MAPK/ERK signaling include those in MET, ALK, RET, NTRK1, and NTRK3 [1]. MET expression was found to be significantly higher in metastases as compared to primary melanomas in a genomic hybridization study [35]. MET also promotes the expression of MMP2, which has been associated with greater metastatic risk and worse prognosis [36].

### 2.4. KIT

KIT encodes the receptor tyrosine kinase c-KIT that is involved in various signaling pathways in melanoma including melanocyte development, MAPK/ERK signaling, PI3K/AKT signaling, and MITF upregulation [5,7]. It is worth noting that 28% of high-CSD, 36% of acral, and 39% of mucosal melanomas were found to have KIT copy number gains or mutations [37]. Interestingly, while KIT expression is initially upregulated in early melanoma, it is often markedly downregulated in later stage disease [38,39,40]. One study found that KIT expression is significantly diminished in melanoma patients with nodal metastases as compared to those without metastasis [38]. Another recent study using the Cancer Genome Atlas (TCGA) and Gene Expression Omnibus (GEO) databases found that KIT expression is significantly lower in metastases as compared to primary melanomas [41]. The disparate expression of KIT in early and late stage disease suggests an important early role in oncogenesis that is subsequently overridden by other drivers [38,39,40,41].

### 2.5. RB Pathway (CDKN2A)

The proteins p14 and p16 (also known as ARF and INK4A respectively) act as tumor suppressors in many cancers including melanoma. Both of these proteins are encoded by the CDKN2A locus [7]. p14 functions as a tumor suppressor by protecting p53 from MDM2-mediated degradation, therefore preventing cell cycle progression [5]. p16 is a cyclin-dependent kinase inhibitor and functions as a tumor suppressor by inhibiting CDK4 and CDK6’s RB-mediated roles in cell cycle progression [5]. In order for CDK4 and CDK6 to exert their functions in cell cycle progression they must bind a D-type cyclin, such as cyclin D1, encoded by CCND1 [5]. CDKN2A is the most commonly altered gene in familial melanoma. and 9p chromosomal losses affecting it have been found in acral and Spitz melanomas [1,42]. Inactivation of CDKN2A is present in up to 90% of all melanomas and usually occurs when a melanoma becomes invasive [7,14]. Loss of function mutations are known to increase in frequency in melanoma metastasis and were found in more than 75% of metastases in the TCGA cohort [6,43]. Bi-allelic loss of CDKN2A is known to significantly increase expression of the transcription factor BRN2, which has been shown to promote metastasis [44]. Other alterations in the RB pathway include those affecting the oncogene CDK4, tumor suppressor TP53, and oncogene CCND1 [14,42,45].

### 2.6. TERT

Certain mutations in the promoter region of TERT result in increased telomerase activity and cell replicative potential [46]. These mutations occur in approximately 50% of cutaneous melanomas, excluding acral and mucosal melanomas where it occurs in 30–41% and 8% of them, respectively [45,47,48,49]. These mutations have also been seen in a subset of familial melanomas and in Spitz melanomas [1,5]. TERT promoter variants often demonstrate a UV signature and are present in most intermediate and melanoma in-situ lesions [1,14]. They have been found to correlate with rapid growth, thicker Breslow depth, and other poor prognostic characteristics for metastasis [50,51]. There is a higher frequency of TERT promoter mutations in melanoma metastases and TERT amplification markedly increases in acral lentiginous melanoma metastasis [5,52]. These data highlight the important role of TERT in the metastatic progression of melanoma [5,50,51,52].

### 2.7. Gαq Pathway

Driving mutations in the Gαq pathway are implicated in uveal melanomas and melanomas arising in or resembling blue nevi (so-called “malignant blue nevi”) [1,53]. The Gαq pathway activates PKC through phospholipase C [7]. PKC then induces MAPK/ERK and YAP signaling [7]. GNAQ and GNA11 are both G protein α subunit family members and more than 80% of uveal and blue-nevus-like melanomas were found to have activating mutations [54,55]. Mutations in GNAQ and GNA11 are generally mutually exclusive and both constitutively activate Gαq signaling [54,55]. Alterations in CYSLTR2 also upregulate Gαq signaling and are present in this subset of melanoma [56]. Less common driving mutations in PLCB4, a phospholipase C enzyme that regulates PKC, occur mutually exclusively to GNAQ, GNA11, and CYSLTR2 alterations [53,57,58].

### 2.8. BAP1, EIF1AX, and SF3B1

BRCA-1 associated protein (BAP1) is a tumor suppressor involved in melanocyte proliferation, differentiation, and DNA repair [5]. Loss of function mutations in BAP1, gain of function mutations in related EIF1AX and SF3B1, and chromosomal 8q gains have been identified as secondary driver mutations in uveal melanomas and are present in a subset of malignant blue nevi [53,58,59]. Up to 40% of uveal melanomas and 17% of malignant blue nevi contain mutations in BAP1 [58,60]. BAP1 germline mutations have been suggested to predispose patients to developing uveal melanoma, cutaneous melanoma, and other malignancies, including mesothelioma and renal cell carcinoma [5]. Mutations in BAP1 are generally mutually exclusive with mutations in SF3B1 and EIFAAX, though exceptions have been identified [53]. SF3B1 mutations are present in approximately 25% of malignant blue nevi and 15% of uveal melanomas while EIF1AX mutations are present in approximately 8% of malignant blue nevi and 24% of uveal melanomas [59,61]. Melanomas with mutations in SF3B1 and EIF1AX are associated with better prognostic outcomes and a lower probability of metastasis than those with mutations in BAP1 [53]. Singh et al. found that co-expression of p52 and RelB was seen more frequently in uveal melanoma metastases and that BAP1 loss promoted their expression [62].

The aforementioned pathways are summarized below in Figure 1.

## 3. Genetics and Transcriptomics of Melanoma Metastasis

Melanoma is now known to have genetically and epigenetically heterogenous clonal populations among both primary and metastatic tumors [63,64]. Most melanoma tumors are recognized to contain heterogenous cell populations with only small subpopulations that have metastatic potential [65]. Melanoma metastases were initially thought to disseminate from a primary tumor to regional lymph nodes and then to distant sites in a serial fashion [9]. However, this model has been questioned since circulating tumor cells can be present before regional or any metastasis at all, and there are patterns suggesting parallel progression in phylogenetic analyses [9]. Shain and Bastian propose that reseeding could occur over time from multiple cells to form melanoma metastases [9]. This would explain why regional metastases are often larger and appear to occur earlier. With reseeding, there is an increased probability for disseminating cells to land in nearer sites to the primary tumor as compared to distant ones, therefore resulting in a larger size [9]. It is well known that the risk of cutaneous melanoma metastasis correlates with Breslow depth of invasion and ulceration of the primary lesion, but a concrete set of metastatic molecular changes have yet to be elucidated [7]. Recently, a common metastatic evolution pathway has been elucidated for uveal melanoma and new research has shed light on the evolution of cutaneous metastatic melanoma, as illustrated in Table 1 and Figure 2. We will discuss the genetic evolution of metastasis in uveal and cutaneous melanomas, recent findings in the realm of transcriptomics, and the effects of the tumor microenvironment on melanoma metastasis.

### 3.1. Pathways of Uveal Melanoma Metastasis

A recent retrospective study by Shain et al. elucidated a common genomic evolution pathway that applied to majority of their metastatic uveal melanomas [53]. As mentioned before, Gαq pathway mutations are the earliest mutations to undergo selection, followed by gain of chromosomal arm 8q and secondary driver mutations in BAP1, SF3B1, or EIF1AX [53]. GNAQ loss of heterozygosity, additional chromatin remodeling mutations, and further ramp-up of 8q copy number occur at later points in the metastatic progression cascade [53]. At intermediate points in the evolutionary pathway, copy number alterations affecting chromosomes 16q, 8p, 1p, 6p, and 6q undergo selection [53]. Chromosome 8q gain and 3, 8p, and 16q loss together were found to be 85.9% predictive of uveal melanoma liver metastasis [24]. The tertiary driver mutations identified by Shain et al. include loss of function of CDKN2A, PBRM1, PIK3R2, and PTEN, gain of function of EZH2, PIK3CA, and MED12, and loss of heterozygosity in the GNAQ gain of function mutation [53]. Though this is the usual pathway of metastatic progression, there were 3 cases in their cohort of 35 where metastatic dissemination preceded 8q gain and 2 cases where metastatic dissemination potentially preceded biallelic loss of BAP1 [53]. This leaves room for metastatic dissemination to occur before the uveal melanoma has developed the full set of mutations [53].

### 3.2. Pathways of Cutaneous Melanoma Metastasis

Compared to uveal melanoma, the genomic evolution of metastasis in cutaneous melanoma is less clear. Soo et al. propose that the genetic drivers of melanomagenesis (e.g., BRAF, NRAS, and KIT) are first acquired, followed by RB pathway changes [66]. As the cutaneous melanoma progresses from a radial growth phase (RGP) to a vertical growth phase (VGP), mutations inhibiting apoptosis (e.g., TP53 or PTEN loss) emerge [66]. TERT promoter mutations then occur, conferring metastatic potential [66]. Birkeland et al. found that whole genome duplication (WGD) then occurs prior to most copy number alterations (CNAs) [67]. This is consistent with the recent findings of Vergara et al. that the acquisition of CNAs and aneuploidy due to both WGD and loss of chromosomes and chromosome arms are characteristic of metastatic cutaneous melanoma [68]. That study also elucidated that allele imbalances tended to occur on specific chromosomes. The authors hypothesized that with those genomic changes, specific alleles were being selected for and against [68]. In contrast with the role acquiring genetic variants plays in early melanomagenesis, mutational accumulation deviates from being UV-induced and is very low during metastasis [67,68]. Metastasis-specific genetic variants in cutaneous melanoma have yet to be elucidated in detail, but several genes have recently been implicated.

#### 3.2.1. Novel AKT Pathway Metastasis Genes

PHLPP1 inhibits melanoma metastasis through its phosphatase activity, repressing AKT2 and to a lesser extent AKT3 [69]. Reduced or lost expression of PHLPP1 was found in 71.4% of metastatic melanoma cell lines and was associated with an worse prognosis in one cohort [69].

#### 3.2.2. Sex-Linked Metastasis Genes

Another study discovered that loss of DDX3X (DEAD-Box Helicase 3 X-Linked) is associated with metastasis formation and decreased distant metastasis-free survival [70]. DDX3X was found to regulate MITF protein levels and mutations were present in 5.8% of their melanoma cohort [70]. Females with melanoma are known to have a survival advantage, a lower risk of progression, and a decreased probability of nodal and visceral metastases [12,71]. One study found that the median time to metastasis was approximately 7 months greater in female patients as compared to males [72]. DDX3X may play a role in this disparity and perhaps future studies will identify other sex-chromosome-specific genes affecting survival and melanoma metastasis [70].

#### 3.2.3. Germline Genes and Metastasis

The germline APOE4 variant was recently discovered to reduce progression and improve survival as compared to APOE2, despite the deleterious effect of APOE4 in Alzheimer’s disease development [73]. This study indicates the propensity for future research to identify other germline variants that may be used as prognostic melanoma biomarkers.

#### 3.2.4. Database Candidate Metastasis Genes

A bioinformatic study using the TCGA and GEO databases identified nine candidate genes as being differentially expressed among primary and metastatic melanomas [74]. Their increased expression was also associated with worse prognosis in the TCGA SKCM cohort [74]. The previously identified genes in melanoma progression (AURKA, BUB1, and PRKCA) were also candidate genes in this study [74,75,76,77]. Another recent study using those databases as well as a transcriptome analysis of a genetically engineered mouse model (GEMM) identified 43 genes that could accurately predict patient prognoses and were upregulated in stage III/IV melanoma samples as compared to stage I/II ones [78]. From this study, GULP1, DAB2, P4HA2, and KDELR3 had their roles in promoting metastasis validated by siRNA knockdown [78]. A mechanistic analysis was further performed on KDELR3 and it was found to relieve ER stress in melanoma, an important function in the survival of metastases [78]. Several other recent studies have also used the TCGA and GEO databases to identify differentially expressed candidate genes in melanoma metastases [79,80]. They include those known to function in desmosomes, the keratinocyte cornified envelope, UV protection, leukocyte chemotaxis, WNT/β-catenin signaling, MAPK/ERK signaling, PI3K/AKT signaling, TGF-β signaling, VEGF signaling, and EMT-like program signaling [41,79,80].

### 3.3. Transcriptomics of Melanoma Metastasis

#### 3.3.1. Transcription and Translation Factors

Transcription and translation factors regulate many important pathways in melanoma metastasis. EMT-like phenotype switching is important for metastatic melanoma progression, and it is orchestrated by MITF, AXL, and EMT-inducing transcription factors (EMT-TFs), such as TWIST, ZEB, and SLUG [81]. Melanocyte inducing transcription factor (MITF) (also known as microphthalmia-associated transcription factor) physiologically plays important roles in melanoblast survival, melanocyte development, and melanosome export. AXL is a receptor tyrosine kinase that is involved in inflammatory processes [81,82]. The high ZEB1/TWIST1, low MITF, high AXL state promotes an invasive, dedifferentiated phenotype in melanoma while the high ZEB2/SLUG, high MITF, low AXL state promotes an anti-invasive, proliferative, differentiated phenotype [81,83]. This is similar to the EMT/MET balancing observed in carcinomas and these factors likely act to promote different phenotypes to adapt to the specific stage and context of the melanoma [81]. A recent study demonstrated that activation of the invasive high ZEB1/AXL program through downregulation of SMAD7 also maintained proliferative ability and high MITF expression [84]. Patients with low SMAD7 expression in that study were found to have significantly decreased overall survival as compared to those with high expression [84]. Endothelin 1 (EDN1) has recently been identified as a regulator of phenotype switching heterogeneity. Through the GPCR endothelin receptor B (EDNRB), EDN1 supports MITF high populations while through endothelin receptor A (EDNRA), it supports AXL high populations [85]. The GPCR melanocortin receptor 1 (MC1R) is also known to regulate and increase MITF levels. Interestingly, it is rarely mutated in melanomas, though germline mutations predispose individuals to developing melanoma [63,86]. Tumors can contain cells expressing both high MITF and low MITF states. Rather than two distinct phenotypes, melanoma remains on a heterogenous spectrum of invasive and proliferative states, which allows for optimization to context and immune evasion [83]. This tumoral heterogeneity contributes significantly to metastasis and patient survival.

Several studies have demonstrated that factors interacting with phenotype switching proteins significantly affect melanoma metastasis progression and heterogeneity. Pietrobono et al. implicated the oncogenic SOX2-GLI1 transcriptional complex as having a role in metastasis through the induction of ST3GAL1 and subsequent dimerization and activation of AXL [87]. Though overexpression of ST3GAL1 did not result in increased ability to metastasize, ST3GAL1 is necessary for melanoma metastasis [87]. The western blot also elucidated that ST3GAL1, SOX2, GLI1, and AXL all have higher expression in metastatic melanomas as compared to primary melanomas [87]. SOX2 has been identified as an independent prognostic indicator for poor survival [88]. Another study found that the transcriptional regulator YAP was necessary and sufficient for melanoma invasion in vitro and was able to promote metastasis while impeding the growth of primary tumors in vivo [89]. This is consistent with YAP promoting ZEB1 expression [83]. YAP/TAZ/TEAD axis signaling is known to promote the low MITF, invasive, antiproliferative state, while the YAP/TAZ/PAX3 axis promotes a higher expression of MITF [82,90]. TGF-β signaling favors the invasive, antiproliferative YAP axis by activating the SMAD/YAP/TAZ/TEAD cascade and inhibiting PAX3, which therefore hinders YAP binding to the MITF promoter [82]. High YAP/TAZ expression has been associated with significant reduction in post-operative survival for melanoma patients [91]. YAP overexpression was also associated with anoikis resistance, which could contribute to its role in melanoma metastasis [92]. The transcription factor BRN2 has been shown to contribute to melanoma metastasis and confer anoikis resistance as well [93]. BRN2 is known to transcriptionally repress MITF and potentially contribute to the high mutational burden of melanoma through promoting the more error-prone non-homologous end-joining DNA repair system [93]. However, a recent study demonstrated in vivo that BRN2 haplo-insufficiency promoted melanomagenesis and metastasis, though metastatic colonization was less efficient [94]. Transcriptional repression of MITF by BRN2 is thought to occur through its downstream effector NFIB and the subsequent upregulation of EZH2 [95]. EZH2 has been found to promote melanoma metastasis by enhancing WNT/β-catenin signaling through the silencing of tumor suppressor genes that maintain the integrity of primary cilia [96].

The translation-affecting proteins eIF4E and sequestosome 1 have also been implicated in metastasis progression. Carter et al. found that reduced survival and risk of metastasis are both correlated with increased eIF4E and phospho-eIF4E expression [97]. Phospho-eIF4E disproportionally promotes the translation of mRNAs associated with cancer progression, such as CCND1 and VEGF [97]. Sequestosome 1 also promotes the translation of prometastatic mRNAs, including FERMT2 [98,99]. It stabilizes these mRNAs via interactions with RNA-binding proteins and IGF2BP1 [98]. Though mRNA expression profiles play a significant role in melanoma progression, little is known about the role of RNA binding proteins in metastasis. Future research should continue to explore the role RNA binding proteins and other related factors have in the metastatic progression of melanoma.

#### 3.3.2. Non-Coding RNA

Recent research has indicated the potential role that long non-coding RNA (lncRNA) and microRNA play in melanoma metastasis and their utility as biomarkers. lncRNAs tend to promote melanoma metastasis through increasing the expression of a transcript by acting as a decoy for its inhibitory microRNA. The lncRNAs that have been associated with melanoma metastasis and their corresponding microRNA and upregulated transcripts include KCNQ1OT1/miR-153/MET [100], LINC00518/miR-204-5p/AP1S2 [101], LINC00520/miR-125b-5p/EIF5A2 [102], and UCA1/hsa-miR-125b-1/AKT1 [103]. LINC00518 and LINC00520 were each found to be independent risk factors for metastatic melanoma patient survival and the network motif UCA1/hsa-miR-125b-1/AKT1 has an 88% chance of correctly identifying a TCGA sample as metastatic melanoma [101,102,103]. These studies demonstrate not only how lncRNA and miRNAs function in melanoma metastasis but also their potential to be prognostic and diagnostic biomarkers. Stark et al. found that a panel of seven serum microRNAs were able to detect metastatic melanoma with a high sensitivity (93%) and specificity (≥82%) [104]. This panel performed better at predicting overall survival, melanoma progression, and recurrence than serum LDH and S100B [104]. Armand–Labit et al. also found that plasma miR-1246 and miR-185 had a high sensitivity (90.5%) and specificity (89.1%) at identifying metastatic melanoma [105]. A small study found 25 miRNAs that were differentially expressed between pre and post-surgical metastatic melanoma plasma samples [106]. There is great potential and utility for lncRNA and miRNA, particularly non-invasive plasma miRNA, to be used as new biomarkers of melanoma metastasis.

### 3.4. Tumor Microenvironment

Though subclinical melanoma metastases have been found in many parts of the body, clinically detectable distant metastases are usually limited to skin, lung, brain, liver, bone, and intestine [65]. This suggests that tumor microenvironments play a significant role in melanoma metastasis. Further supporting this is the observation that some patients developed melanoma metastases after receiving organ transplants from supposedly disease-free individuals with a history of melanoma, ostensibly resulting from reactivation of the donor’s dormant metastatic disease [65]. A recent study by Tirosh et al. using single-cell RNA sequencing to analyze the transcriptome of melanoma tumors found that the tumor microenvironment is shaped by the intra- and interindividual spatial, functional, and genomic heterogeneity of melanoma and the associated tumor components [64]. A later study by Thrane et al. using in situ transcriptome analysis showed intra- and intertumoral gene expression heterogeneity in lymph node metastases. They also found that lymphoid tissue gene expression programs were potentially influenced by their distance from tumor cell clusters. In lymphoid tissues far away from tumor cell areas expression of the immune-related gene CD74 was observed while in tissues in close proximity to tumor cell areas, expression of the immune-related gene IGLL5 was seen. In the tumor cell cluster, PMEL and SPP1 were overexpressed [107]. It is thought that components of the adaptive immune system inhibit metastasis while innate immune system cells, such as macrophages, assist in mediating metastasis [65]. Overexpression of αvβ3/αIIbβ3 integrins on melanoma cells, which is mediated by EMT-TFs, is known to be associated with metastasis and the switch from RGP to VGP [81,108]. αvβ3-integrin has also been found to mediate extracellular matrix degradation and regulate the expression PD-L1, therefore promoting immune evasion [81]. Autocrine motility factor (AMF), a cytokine that is secreted from tumor cells, has been shown by Tímár et al. to be linked with in vivo spontaneous metastatic potential and greater β3 integrin expression [109]. CD44, a cell surface adhesion receptor that participates in lymphocyte activation as well as other immune functions, has also been linked with melanoma metastasis. Döme et al. found that the five year survival of patients with melanomas greater than 1mm thickness was significantly lower for those positive for the CD44v3 splice variant than those that were negative [110]. Other metastasis-associated factors on melanoma cells that affect the tumor microenvironment include MCAM/MUC18, L1-CAM, α4β1-integrin, ECM remodeling molecule FN1, and glypican-6 [81]. Melanoma is known to evade the immune system by increasing expression of PD-L1 (ligand for the T cell PD-1 receptor) on melanoma cells and CTLA-4 (an immune suppressive protein) on T cells. Their increased expression promotes immunosuppressive activity, such T cell anergy, Treg differentiation, and tumor infiltrating lymphocyte apoptosis. Progression is associated with the presence of Treg cells due to their ability to help melanoma avoid immune surveillance through the secretion of cytokines and chemokines (CXC) with immunosuppressive actions, such as IL-10, IL-35, and TGF-β. Melanoma also suppresses the immune system by expressing immune inhibiting miRNAs, depriving T cells of arginine and tryptophan, inhibiting natural killer cell and macrophage function through high levels of adenosine, inhibiting natural killer cell activity and APC immunogenicity through high levels of kinurenine, and the secretion of exosomes that modulate the tumor microenvironment. The immune evasion strategies have been covered in greater detail in previous reviews [111]. NRAS, KIT, EGFR, MET, RAB27A, and other pro-progression proteins have all been found within exosomes released by melanomas [112]. Melanoma exosomes are able to promote invasion, angiogenesis, and metastasis in receiving tumor cells [112]. They are also able to create premetastatic niches in the body [112]. Melanoma metastases are known to favor colonizing areas with higher exosomes and sentinel lymph nodes were found to have very high levels [112]. Lymph metastases are known to often precede the discovery of those in the blood and distant sites. A recent finding is that lymph prevents ferroptosis and reduces oxidative stress in metastasizing melanoma cells [113]. This is due to the reduced levels of free iron and elevated levels of glutathione and oleic acid in the lymph as compared to the blood [113]. They also discovered that melanoma cells that were previously exposed to lymph survived better in blood than those that had no exposure since they incorporated the antioxidants from the lymph [113]. Melanoma also has the ability to perform platelet and vasogenic mimicry. Platelet mimicry is characterized by the expression of the megakaryoctic factors and the GPCR PAR1 (protease activating receptor 1) is associated with this [114]. Future studies should continue elucidating novel factors affecting the tumor microenvironment and promoting metastasis.

### 3.5. Melanoma Progression during Targeted and Immune Therapies

The introduction of targeted and immunotherapies for melanoma has resulted in great improvements in patient survival. The identification of melanoma driver mutations in the MAPK/ERK pathway resulted in the development of BRAF small molecule inhibitors (vemurafenib and dabrafenib) and MEK small molecule inhibitors (trametinib and cobimetinib). Immune checkpoint inhibitors are also commonly used and include the cytotoxic T-lymphocyte-associated antigen 4 (CTLA-4) inhibitor (ipilimumab) and the programmed cell death protein 1 (PD-1) inhibitors (pembrolizumab and nivolumab). Despite these advances in treatment, long-term success is still rare for late stage melanoma patients due to the development of drug resistance [82].

There is a great amount of genetic and epigenetic heterogeneity within melanoma tumors comprised of different clonal populations [63]. Therapeutic resistance could therefore arise from “Darwinian” selection of subclones that are able to survive treatment or from a “Lamarckian” selective pressure for tumors to acquire changes that allow them to persist [23,82]. Melanomas with NRAS-mutant cells and BRAF-mutant cells have greater resistance to BRAF inhibitors, likely due to selection of NRAS mutant populations. HOXD8 and RAC1 mutations have also been shown to give melanomas primary resistance to targeted therapies [63].

In patients taking BRAF and MEK inhibitors, MAPK/ERK pathway reactivation occurs in up to 80% of tumors through mechanisms such as alterations in MEK, NRAS, PTEN and NF1, and BRAF allele amplification and/or splice variants [63,115]. There can also be increased expression of alternate MAPK/ERK pathway activators, such as CRAF (C Rapidly Accelerated Fibrosarcoma Serine/Threonine Protein Kinase) and MAP3K8 (Mitogen-Activated Protein Kinase Kinase Kinase 8) [116,117]. Outside of restoring MAPK/ERK signaling, the PI3K/AKT pathway is commonly activated to compensate [63]. Note that multiple different mechanisms of resistance can occur in synchronous tumors of the same melanoma. One study found that 20% of melanoma patients after initiation of BRAF-inhibitor therapy developed two or more resistance mechanisms [63,118].

Drug resistance is also mediated by the tumor microenvironment and through transcriptomic changes. Stromal cell secretion of growth factors has been shown to promote MAPK/ERK and PI3K/AKT pathways [119,120]. Other reviews have discussed in more detail the ways the tumor microenvironment promotes resistance [121]. Multiple miRNAs have also been implicated in drug resistance [122,123]. One example is the upregulation of miR-211-5p and miR-204-5p via MITF and STAT3 (signal transduce and activator of transcription 3), which has been shown to reactivate MAPK/ERK and PI3K/AKT signaling, conferring therapeutic resistance [124]. Transcriptionally-regulated phenotype switching plays a major role in the ability of melanoma to evade therapies. The heterogeneity of melanoma having cells in both MITF high and low states and their ability to transition from one state to another is thought to confer drug resistance. Cells in the MITF low, AXL high, invasive, dedifferentiated state are slow-proliferating, and this allows them to avoid current therapies which primarily target fast-proliferating cells. In addition, AXL has been shown to promote the MAPK/ERK and PI3K/AKT pathways and this may also contribute to therapeutic evasion of these cells [82]. As a result, resistance to several targeted therapies is predicted by a low MITF/AXL ratio [125].

One study found that that using an AXL antibody drug conjugate in combination with a MAPK inhibitor targeted both slow and fast proliferating cells and demonstrated improved treatment results [125]. Another study found that endothelin receptor antagonists suppressed high AXL tumor populations and re-sensitized cells to BRAF inhibitors [85]. Outside of targeting AXL high populations, Kenessey et al. found a greater benefit of treatment from combining BRAF inhibitors with epidermal growth factor tyrosine kinase inhibitors [126]. These studies demonstrate that there is great promise in treating metastatic melanoma with combined therapies against novel therapeutic targets in the future.

## 4. Summary

Recent research has shed new light on the genomic and transcriptomic factors promoting the progression of melanoma metastasis. Though more is now known about the genetic evolution of metastasis, particularly for uveal melanoma, the precise molecular mechanisms are still largely undefined. Future studies should continue utilizing the large databases (e.g., TCGA and GEO) to identify differentially expressed candidate genes, new mechanisms of melanoma metastasis, and novel therapeutic targets. Germline variants could also be considered as novel prognostic biomarkers and their role in melanoma progression should be explored further. lncRNA, microRNA, and non-coding RNAs have shown great promise as diagnostic and prognostic biomarkers, particularly for monitoring disease status. The tumor microenvironment serves as an increasingly promising arena of exploration for identification of prognostic markers and potential therapeutic targets for patients who fail conventional treatments. Combination therapies aimed at overcoming drug resistance show great promise for better prognostic outcomes.

## Figures and Tables

**Figure 1 cancers-14-00123-f001:**
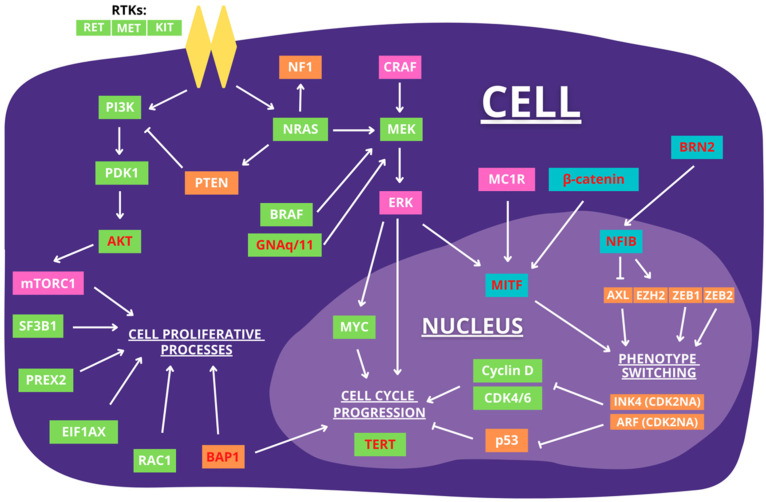
Illustration of pathway components involved in melanoma progression. Orange indicates the protein is silenced, green indicates upregulation, and pink indicates upregulation or downregulation during progression. Gene written in red are implicated to have importance in melanoma metastasis.

**Figure 2 cancers-14-00123-f002:**
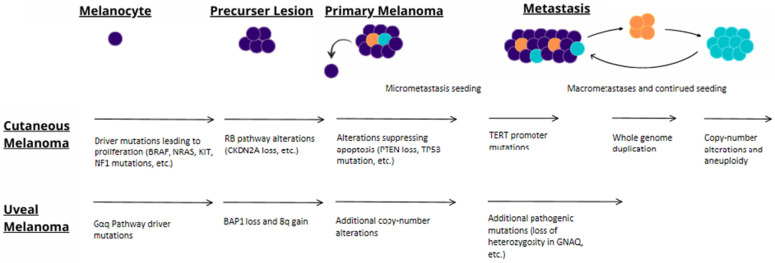
Proposed model of progression to metastasis for cutaneous and uveal melanoma.

**Table 1 cancers-14-00123-t001:** Proposed genomic changes during the progression to melanoma, adapted from the World Health Organization Classification of Skin Tumours, 4th Edition. The color scheme is as follows: gain of function, loss of function, amplification, rearrangement, change of function, promoter mutation.

	Superficial Spreading Melanoma	Lentigo Maligna Melanoma	Desmoplastic Melanoma	Spitz Melanoma	Acral Melanoma	Mucosal Melanoma	Melanoma in Congenital Nevus	Melanoma in Blue Nevus	Uveal Melanoma
**Initiation Mutations**	BRAF, NRAS Rarely: MAP2K1, CTNNB1, APC, BAP1, PRKAR1A, PRKCA	NRAS, BRAF, KIT, NF1	NF1, ERBB2, MAP2K1, MAP3K1, BRAF, EGFR, MET	HRAS, ALK, ROS1, RET, NTRK1, NTRK3, BRAF, MET	KIT, NRAS, BRAF, HRAS, KRAS, NTRK3, ALK, NF1	KIT, NRAS, KRAS, BRAF	NRAS, BRAF	GNAQ, GNA11, CYSLTR2	GNAQ, GNA11, CYSLTR2, PLCB4
**Malignant Transformation** **Mutations**	TERT, CDKN2A, TP53, PTEN	TERT, CDKN2A, TP53, PTEN, RAC1	TERT, NFKBIE, NRAS, PIK3CA, PTPN11	CDKN2A	CDKN2A, TERT, CCND1, GAB2	NF1, CDKN2A, SF3B1, CCND1, CDK4, MDM2		BAP1, EIF1AX, SF3B1	BAP1, EIF1AX, SF3B1
**Metastasis Progression Genomic Changes**	Whole genome duplication, followed by further copy number alterations and aneuploidy	Whole genome duplication, followed by further copy number alterations and aneuploidy	Whole genome duplication, followed by further copy number alterations and aneuploidy						Additional copy number alterations and mutations (GNAQ LOH, etc.)

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
