# Peer review of "Genomic and Transcriptomic Underpinnings of Melanoma Genesis, Progression, and Metastasis"

_cancers, 2021, doi:10.3390/cancers14010123_

Round 1

Reviewer 1 Report

Genomic and transcriptomic underpinnings of melanoma 2 genesis, progression, and metastasis,

By  Olga S Cherepakhin, et al.

________________________________________________________

The current paper focuses on a gene-, and transcript review on the field of Melanoma that is of mandatory importance to the society as the rate of Melanoma patients are increasing considerably, and especially in the 20-40 age group, which is alarming.

The paper is well written and structured. I would like to propose a more reader friendly structure with more than 2 Figures and a Layout that is more appealing.

I have also added some comments below that relates to the parts and details within the manuscript.

  • Affiliations to the authors needs to be added
  • This reviewer finds it remarkable that the studies by the Timars team at Semmelweis are not referenced at all….- as these have manifested important landmarks in Melanoma, they need to be added.
  • Don’t repeat exactly the same sentence; “Melanoma is a deadly skin cancer with rapidly increasing incidence worldwide.”
  • I miss out on parts that relates to the attribution of phenotype switching, that recently has been published on in several studies
  • I would like the authors to discuss in more detail the aspect of tumor heterogeneity and its link as precursors to invasion and metastasis
  • I am missing out in a table with a Mutational Melanoma Landscapes of that encorporates the Somatic Melanomas as well
  • The aspect of epithelial to mesenchymal transitions needs to be addressed to a more extent
  • I miss out on a part that relates to what impact epigenomic alterations are crucial in mortality and that contributes to the heterogeneity within the tumor
  • Would be of interest to the readers to have an overview of the Drugs available for treatment, as well as the Drug Impact mechanisms being utitlized in patient treatments
  • Could you elaborate on additional G-protein coupled receptors (GPCRs), that would include FZD, such as MC1R, EDNR), C-X-C and CXCR) and PAR1 activated in metastatic melanoma progression….
  • What about TGF-B and the SMAD pathway activations..?
  • I would like to see more on the Tumor Complexity, that relates to the Tumor Progression and the build on Drug Resistance
  • What about the latest data coming up on Tumor complexity and the knowledge build on with cancer cell clones in relation to stromal cells and immune cells – in the battle taking place within the tumor microenvironments and compartments in recent studies also by; Tirosh; and Thrane..et al

Author Response

Please see combined comments and revisions in this cover letter, thank you!

Comments by Reviewer #1

  • Affiliations to the authors need to be added

Author affiliations have been added on the first page.

  • This reviewer finds it remarkable that the studies by the Timars team at Semmelweis are not referenced at all as these have manifested important landmarks in Melanoma, they need to be added.

We fully agree, this was an oversight by us. Eight papers authored by Timar’s team at Semmelweis have been added to the manuscript and are visible in the References section on pages 14-25. The papers cited are as follows:

  • Doma V, Kárpáti S, Rásó E, Barbai T, Tímár J. Dynamic and unpredictable changes in mutant allele fractions of BRAF and NRAS during visceral progression of cutaneous malignant melanoma. BMC Cancer. 2019 Aug 7;19(1):786. doi: 10.1186/s12885-019-5990-9. Erratum in: BMC Cancer. 2019 Aug 29;19(1):853. PMID: 31391014; PMCID: PMC6686548.
  • Tímár J, Vizkeleti L, Doma V, Barbai T, Rásó E. Genetic progression of malignant melanoma. Cancer Metastasis Rev. 2016 Mar;35(1):93-107. doi: 10.1007/s10555-016-9613-5. PMID: 26970965.
  • Ladányi A, Tímár J, Bocsi J, Tóvári J, Lapis K. Sex-dependent liver metastasis of human melanoma lines in SCID mice. Melanoma Res. 1995 Apr;5(2):83-6. doi: 10.1097/00008390-199504000-00002. PMID: 7620343.
  • Trikha M, Timar J, Zacharek A, Nemeth JA, Cai Y, Dome B, Somlai B, Raso E, Ladanyi A, Honn KV. Role for beta3 integrins in human melanoma growth and survival. Int J Cancer. 2002 Sep 10;101(2):156-67. doi: 10.1002/ijc.10521. PMID: 12209993
  • Tímár J, Rásó E, Döme B, Ladányi A, Bánfalvi T, Gilde K, Raz A. Expression and function of the AMF receptor by human melanoma in experimental and clinical systems. Clin Exp Metastasis. 2002;19(3):225-32. doi: 10.1023/a:1015595708241. PMID: 12067203.
  • Döme B, Somlai B, Ladányi A, Fazekas K, Zöller M, Tímár J. Expression of CD44v3 splice variant is associated with the visceral metastatic phenotype of human melanoma. Virchows Arch. 2001 Nov;439(5):628-35. doi: 10.1007/s004280100451. PMID: 11764382.
  • Timar, J., Barbai, T., GyÅ‘rffy, B., & Rásó, E. Understanding melanoma progression by gene expression signatures. In U. Pfeffer (Ed.), Cancer genomics: Molecular classification, prognosis and response prediction. Dordrecht: Springer. 2013. 47–79. doi: 10.1007/978-94-007-5842-1
  • Kenessey I, Kramer Z, István L, Cserepes MT, Garay T, Hegedűs B, Dobos J, Tímár J, Tóvári J. Inhibition of epidermal growth factor receptor improves antitumor efficacy of vemurafenib in BRAF-mutant human melanoma in preclinical model. Melanoma Res. 2018 Dec;28(6):536-546. doi: 10.1097/CMR.0000000000000488. PMID: 30124539.

  • Don’t repeat the exact same sentence; “Melanoma is a deadly skin cancer with rapidly increasing incidence worldwide.”

The sentence has now been changed to “Melanoma is a skin cancer with a high mortality and dramatically rising presence worldwide” on page 1 in the Simple Summary.

  • I miss out on parts that relates to the attribution of phenotype switching, that recently has been published on in several studies. The aspect of epithelial to mesenchymal transitions needs to be addressed to a greater extent. I miss out on a part that relates to what impact epigenomic alterations are crucial in mortality and that contributes to the heterogeneity within the tumor.

We discussed phenotype switching, EMT transitions, and the effect of these epigenomic alterations on mortality in more detail in both the Transcription and Translation Factors section on pages 9-10, and in the added Melanoma Progression During Targeted and Immune Therapies section on page 13.

Page 9:

“Transcription and translation factors regulate many important pathways in melanoma metastasis. EMT-like phenotype switching is important for metastatic melanoma progres-sion, and it is orchestrated by MITF, AXL, and EMT-inducing transcription factors (EMT-TFs), such as TWIST, ZEB, and SLUG. [81] Melanocyte inducing transcription factor (MITF), also known as microphthalmia-associated transcription factor physiologically plays important roles in melanoblast survival, melanocyte development, and melanosome export. AXL is a receptor tyrosine kinase that is involved in inflammatory processes. [82][81]The high ZEB1/TWIST1, low MITF, high AXL state promotes an invasive, dediffer-entiated phenotype in melanoma while the high ZEB2/SLUG, high MITF, low AXL state promotes an anti-invasive, proliferative, differentiated phenotype. [81][83] This is similar to the EMT/MET balancing observed in carcinomas and these factors likely act to promote different phenotypes to adapt to the specific stage and context of the melanoma. [81] [83]A recent study demonstrated that activation of the invasive high ZEB1/AXL program through downregulation of SMAD7 also maintained proliferative ability and high MITF expression. [84] Patients with low SMAD7 expression in that study were found to have significantly decreased overall survival as compared to those with high expression. [84] Endothelin 1 (EDN1) has recently been identified as a regulator of phenotype switching heterogeneity. Through the GPCR endothelin receptor B (EDNRB), EDN1 supports MITF high populations while through endothelin receptor A (EDNRA), it supports AXL high populations.[85] The GPCR melanocortin receptor 1 (MC1R) is also known to regulate and increase MITF levels. Interestingly, it is rarely mutated in melanomas, though germline mutations predispose individuals to developing melanoma.[63][86] Tumors can contain cells expressing both high MITF and low MITF states. Rather than two distinct phenotypes, melanoma remains on a heterogenous spectrum of invasive and proliferative states, which allows for optimization to context and immune evasion. [83] This tumoral heterogeneity contributes significantly to metastasis and patient survival.”

Page 9:

“Pie-trobono et al implicated the oncogenic SOX2-GLI1 transcriptional complex as having a role in metastasis, through the induction of ST3GAL1 and subsequent dimerization and activation of AXL. [87] Though overexpression of ST3GAL1 did not result in increased ability to metastasize, ST3GAL1 is necessary for melanoma metastasis.”

Page 10:

Another study found that the transcriptional regulator YAP was necessary and sufficient for melanoma invasion in vitro and was able to promote metastasis while impeding the growth of primary tumors in vivo. [91] This is consistent with it promoting ZEB1 expression. [83] YAP/TAZ/TEAD axis signaling is known to promote the low MITF, invasive, antiproliferative state while the YAP/TAZ/PAX3 axis promotes a higher expression of MITF. [89][82] TGF-β signaling favors the invasive, antiproliferative YAP axis by activating the SMAD/YAP/TAZ/TEAD cascade and inhibiting PAX3, which therefore hinders YAP binding to the MITF pro-moter.[82] High YAP/TAZ expression has been associated with significant reduction in post-operative survival for melanoma patients. [90] YAP overexpression was also associ-ated with anoikis resistance, which could contribute to its role in melanoma metastasis. [92]”

Page 13:

“Transcriptionally-regulated phenotype switching plays a major role in the ability of melanoma to evade therapies. The heterogeneity of melanoma having cells in both MITF high and low states and their ability to transition from one state to another is thought to confer drug resistance. Cells in the MITF low, AXL high, invasive, dedifferentiated state are slow-proliferating, and this allows them to avoid current therapies which primarily target fast-proliferating cells. Also, AXL has been shown to promote the MAPK/ERK and PI3K/AKT pathways and this may also contribute to therapeutic evasion of these cells. [82] As a result, resistance to several targeted therapies is predicted by a low MITF/AXL ratio. [125]

One study found that that using an AXL antibody drug conjugate in combination with a MAPK inhibitor targeted both slow and fast proliferating cells and demonstrated improved treatment results. [125] Another study found that endothelin receptor antagonists suppressed high AXL tumor populations and re-sensitized cells to BRAF inhibitors.[85]”

  • I would like the authors to discuss in more detail the aspect of tumor heterogeneity and its link as precursors to invasion and metastasis. I would like to see more on the Tumor Complexity, that relates to the Tumor Progression and the build on Drug Resistance. What about the latest data coming up on Tumor complexity and the knowledge build on with cancer cell clones in relation to stromal cells and immune cells – in the battle taking place within the tumor microenvironments and compartments in recent studies also by; Tirosh; and Thrane..et al

We expanded on the role of tumor heterogeneity and complexity in the Genetics and Transcriptomics of Melanoma Metastasis section on page 6, Transcription and Translation Factors section on page 9, Tumor Microenvironment on page 11, and in the added Melanoma Progression During Targeted and Immune Therapies section on pages 12-13. The studies by Tirosh (64) and Thrane et al (107) are discussed on page 11.

Page 6:

“Melanoma is now known to have genetically and epigenetically heterogenous clonal populations among both primary and metastatic tumors. [63][64] Most melanoma tumors are recognized to contain heterogenous cell populations with only small subpopulations that have metastatic potential. [65] Melanoma metastases were initially thought to dis-seminate from a primary tumor to regional lymph nodes and then to distant sites in a serial fashion. [9] However, this model has been questioned since circulating tumor cells can be present before regional or any metastasis at all, and there are patterns suggesting parallel progression in phylogenetic analyses. [9]”

Page 9:

“Endothelin 1 (EDN1) has recently been identified as a regulator of phenotype switching heterogeneity. Through the GPCR endothelin receptor B (EDNRB), EDN1 supports MITF high populations while through endothelin receptor A (EDNRA), it supports AXL high populations.[85] The GPCR melanocortin receptor 1 (MC1R) is also known to regulate and increase MITF levels. Interestingly, it is rarely mutated in melanomas, though germline mutations predispose individuals to developing melanoma.[63][86] Tumors can contain cells expressing both high MITF and low MITF states. Rather than two distinct phenotypes, melanoma remains on a heterogenous spectrum of invasive and proliferative states, which allows for optimization to context and immune evasion. [83] This tumoral heterogeneity contributes significantly to metastasis and patient survival.

Several studies have demonstrated that factors interacting with phenotype switching proteins significantly affect melanoma metastasis progression and heterogeneity.”

Page 11:

“This suggests that tumor microenvironments play a significant role in melanoma metastasis. Further supporting this is the observation that some patients developed melanoma metastases after receiving organ transplants from supposedly dis-ease-free individuals with a history of melanoma, ostensibly resulting from reactivation of the donor’s dormant metastatic disease. [65] A recent study by Tirosh et al. using single-cell RNA sequencing to analyze the transcriptome of melanoma tumors found that the tumor microenvironment is shaped by the intra- and interindividual spatial, functional, and genomic heterogeneity of melanoma and the associated tumor components. [64] A later study by Thrane et al. using in situ transcriptome analysis showed intra- and inter-tumoral gene expression heterogeneity in lymph node metastases. They also found that lymphoid tissue gene expression programs were potentially influenced by their distance from tumor cell clusters. In lymphoid tissues far away from tumor cell areas expression of the immune-related gene CD74 was observed while in tissues in close proximity to tu-mor cell areas, expression of the immune-related gene IGLL5 was seen. In the tumor cell cluster, PMEL and SPP1 were overexpressed. [107]”

Page 12:

“There is a great amount of genetic and epigenetic heterogeneity within melanoma tumors among different clonal populations. [63] Therapeutic resistance therefore could arise from "Darwinian" selection of subclones that are able to survive treatment or from a "Lamarckian" selective pressure for tumors to acquire changes that allow them to withstand treatment.[23][82] Melanomas which have NRAS-mutant cells in conjunction with BRAF-mutant cells have greater resistance to BRAF inhibitors likely due to selection of NRAS mutant populations. Also specific HOXD8 and RAC1 mutations have also been shown to give melanomas primary resistance to targeted therapies. [63]

In patients taking BRAF and MEK inhibitors, MAPK/ERK pathway reactivation occurs in up to 80% of tumors through mechanisms such as alterations in MEK, NRAS, PTEN and NF1, and BRAF allele amplification and/or splice variants.[115][63] There can also be increased expression of alternate MAPK/ERK pathway activators, such as CRAF (C Rapidly Accelerated Fibrosarcoma Serine/Threonine Protein Kinase) and MAP3K8 (Mitogen-Activated Protein Kinase Kinase Kinase 8).[116][117] Outside of restoring MAPK/ERK signaling, the PI3K/AKT pathway is commonly activated to compensate.[63] Note that multiple different mechanisms of resistance can occur in synchronous tumors of the same melanoma. One study found that 20% of melanoma patients after initiation of BRAF-inhibitor therapy developed two or more resistance mechanisms. [118][63]”

Page 13:

“Transcriptionally-regulated phenotype switching plays a major role in the ability of melanoma to evade therapies. The heterogeneity of melanoma having cells in both MITF high and low states and their ability to transition from one state to another is thought to confer drug resistance. Cells in the MITF low, AXL high, invasive, dedifferentiated state are slow-proliferating, and this allows them to avoid current therapies which primarily target fast-proliferating cells. Also, AXL has been shown to promote the MAPK/ERK and PI3K/AKT pathways and this may also contribute to therapeutic evasion of these cells. [82] As a result, resistance to several targeted therapies is predicted by a low MITF/AXL ratio. [125]”

  • I am missing out on a table with a Mutational Melanoma Landscapes of that incorporates the Somatic Melanomas as well

A mutational melanoma landscapes table has been added to the manuscript on page 9. It discusses the mutational landscape of the 9 different types of melanoma from initiation to metastasis.

  • Would be of interest to the readers to have an overview of the Drugs available for treatment, as well as the Drug Impact mechanisms being utitlized in patient treatments

An overview of the common drugs available for treatment has been added to the newly created Melanoma Progression During Targeted and Immune Therapies section on page 12.

Page 12:

The introduction of targeted and immunotherapies for melanoma has resulted in great improvements in patient survival. The identification of melanoma driver mutations in the MAPK/ERK pathway resulted in the development of BRAF small molecule inhibitors (vemurafenib and dabrafenib) and MEK small molecule inhibitors (trametinib and cobimetinib). Immune checkpoint inhibitors are also commonly used and include the cytotoxic T-lymphocyte-associated antigen 4 (CTLA-4) inhibitor (ipilimumab) and the programmed cell death protein 1 (PD-1) inhibitors, (pembrolizumab and nivolumab). Despite these advances in treatment, long-term success is still rare for late stage melanoma patients due to the development of drug resistance.[82]”

  • Could you elaborate on additional G-protein coupled receptors (GPCRs), that would include FZD, such as MC1R, EDNR), C-X-C and CXCR) and PAR1 activated in metastatic melanoma progression….

We incorporated descriptions of the roles and effectors of these GPCRs in the Transcription and Translation Factors section on page 9 and in the Tumor Microenvironment section on pages 11-12.

Page 9:

“Endothelin 1 (EDN1) has recently been identified as a regulator of phenotype switching heterogeneity. Through the GPCR endothelin receptor B (EDNRB), EDN1 supports MITF high populations while through endothelin receptor A (EDNRA), it supports AXL high populations.[85] The GPCR melanocortin receptor 1 (MC1R) is also known to regulate and increase MITF levels. Interestingly, it is rarely mutated in melanomas, though germline mutations predispose individuals to developing melanoma.[63][86]”

Page 11:

“Progression is associated with the presence of Treg cells due to their ability to help melanoma avoid immune surveillance through the secretion of cyto-kines and chemokines (CXC) with immunosuppressive actions, such as IL-10, IL-35, and TGF-β.”

Page 12:

“Melanoma also has the ability to do platelet and vasogenic mimicry. Platelet mimicry is characterized by the expression of the megakar-yoctic factors and the GPCR PAR1 (protease activating receptor 1) is associated with this. [114]”

  • What about TGF-B and the SMAD pathway activations..?

The TGF-B and SMAD pathway activations are now discussed more in the Transcription and Translation Factors section on pages 9-10, Tumor Microenvironment section on page 11, and in the added Melanoma Progression During Targeted and Immune Therapies section on pages 12-13.

Page 9:

“A recent study demonstrated that activation of the invasive high ZEB1/AXL program through downregulation of SMAD7 also maintained proliferative ability and high MITF expression. [84] Patients with low SMAD7 expression in that study were found to have significantly decreased overall survival as compared to those with high expression. [84]”

Page 10:

“YAP/TAZ/TEAD axis signaling is known to promote the low MITF, invasive, antiproliferative state while the YAP/TAZ/PAX3 axis promotes a higher expression of MITF. [89][82] TGF-β signaling favors the invasive, antiproliferative YAP axis by activating the SMAD/YAP/TAZ/TEAD cascade and inhibiting PAX3, which therefore hinders YAP binding to the MITF pro-moter.[82]”

Page 11:

“Progression is associated with the presence of Treg cells due to their ability to help melanoma avoid immune surveillance through the secretion of cyto-kines and chemokines (CXC) with immunosuppressive actions, such as IL-10, IL-35, and TGF-β.”

Reviewer 2 Report

In this manuscript, the authors summarize the role of molecular pathways, genomic factors, and the tumor microenvironment in the progression from local melanoma to distant disease. The review is well written, and I would recommend this work for publication after addressing some minor issues.

1) For broad audiences, the full names of abbreviations should be presented at their first appearance, such as C-to-T, CC-to-TT, MAPK/ERK, BRAF, RAS, and NF1 in page 2.

2) The formats of figure captions are not uniform.

3) The section of Tumor Microenvironment is too short, since the author has emphasized this section in Abstract and Introduction. Considering this is a hot research area, detailed descriptions of some important examples are necessary.

Author Response

Please see changes and responses in this combined cover letter, thank you!

Comments by Reviewer #2

  • For broad audiences, the full names of abbreviations should be presented at their first appearance, such as C-to-T, CC-to-TT, MAPK/ERK, BRAF, RAS, and NF1 in page 2.

The full names of the abbreviations have now been presented at their first appearance on page 2.

  • The formats of figure captions are not uniform.

The formats of the figure and table captions on pages 6, 7, and 9 are now all uniform.

  • The section of Tumor Microenvironment is too short, since the author has emphasized this section in Abstract and Introduction. Considering this is a hot research area, detailed descriptions of some important examples are necessary.

The Tumor Microenvironment section on pages 11-12, and in the added Melanoma Progression During Targeted and Immune Therapies section on pages 12-13.

Pages 11-12:

“Though subclinical melanoma metastases have been found in many parts of the body, clinically detectable distant metastases are usually limited to skin, lung, brain, liver, bone, and intestine. [65] This suggests that tumor microenvironments play a significant role in melanoma metastasis. Further supporting this is the observation that some patients developed melanoma metastases after receiving organ transplants from supposedly dis-ease-free individuals with a history of melanoma, ostensibly resulting from reactivation of the donor’s dormant metastatic disease. [65] A recent study by Tirosh et al. using single-cell RNA sequencing to analyze the transcriptome of melanoma tumors found that the tumor microenvironment is shaped by the intra- and interindividual spatial, functional, and genomic heterogeneity of melanoma and the associated tumor components. [64] A later study by Thrane et al. using in situ transcriptome analysis showed intra- and inter-tumoral gene expression heterogeneity in lymph node metastases. They also found that lymphoid tissue gene expression programs were potentially influenced by their distance from tumor cell clusters. In lymphoid tissues far away from tumor cell areas expression of the immune-related gene CD74 was observed while in tissues in close proximity to tu-mor cell areas, expression of the immune-related gene IGLL5 was seen. In the tumor cell cluster, PMEL and SPP1 were overexpressed. [107] It is thought that components of the adaptive immune system inhibit metastasis while innate immune system cells, such as macrophages, assist in mediating metastasis. [65] Overexpression of αvβ3/αIIbβ3 integ-rins on melanoma cells, which is mediated by EMT-TFs, is known to be associated with metastasis and the switch from RGP to VGP. [108][81] αvβ3-integrin has also been found to mediate extracellular matrix degradation and regulate the expression PD-L1, therefore promoting immune evasion. [81] Autocrine motility factor (AMF), a cytokine that is se-creted from tumor cells, has been shown by Tímár et al. to be linked with in vivo sponta-neous metastatic potential and greater β3 integrin expression. [109] CD44, a cell surface adhesion receptor that participates in lymphocyte activation as well as other immune functions, has also been linked with melanoma metastasis. Döme et al. found that the five year survival of patients with melanomas greater than 1mm thickness was significantly lower for those positive for the CD44v3 splice variant than those that were negative. [110] Other metastasis-associated factors on melanoma cells that affect the tumor microenvi-ronment include MCAM/MUC18, L1-CAM, α4β1-integrin, ECM remodeling molecule FN1, and glypican-6. [81] Melanoma is known to evade the immune system by increasing expression of PD-L1 (ligand for the T cell PD-1 receptor) on melanoma cells and CTLA-4 (an immune suppressive protein) on T cells. Their increased expression promotes immu-nosuppressive activity, such T cell anergy, Treg differentiation, and tumor infiltrating lymphocyte apoptosis. Progression is associated with the presence of Treg cells due to their ability to help melanoma avoid immune surveillance through the secretion of cyto-kines and chemokines (CXC) with immunosuppressive actions, such as IL-10, IL-35, and TGF-β. Melanoma also suppresses the immune system by expressing immune inhibiting miRNAs, depriving T cells of arginine and tryptophan, inhibiting natural killer cell and macrophage function through high levels of adenosine, inhibiting natural killer cell activ-ity and APC immunogenicity through high levels of kinurenine, and the secretion of exo-somes that modulate the tumor microenvironment . The immune evasion strategies have been covered in greater detail in previous reviews. [111] NRAS, KIT, EGFR, MET, RAB27A and other pro-progression proteins have all been found within exosomes re-leased by melanomas. [112] Melanoma exosomes are able to promote invasion, angiogen-esis, and metastasis in receiving tumor cells. [112] They are also able to create premeta-static niches in the body. [112] Melanoma metastases are known to favor colonizing areas with higher exosomes and sentinel lymph nodes were found to have very high levels.

[112] Lymph metastases are known to often precede the discovery of those in the blood and distant sites. A recent finding is that lymph prevents ferroptosis and reduces oxida-tive stress in metastasizing melanoma cells. [113] This is due to the reduced levels of free iron and elevated levels of glutathione and oleic acid in the lymph as compared to the blood. [113] They also discovered that melanoma cells that were previously exposed to lymph survived better in blood than those that had no exposure since they incorporated the antioxidants from the lymph. [113] Melanoma also has the ability to do platelet and vasogenic mimicry. Platelet mimicry is characterized by the expression of the megakar-yoctic factors and the GPCR PAR1 (protease activating receptor 1) is associated with this. [114] Future studies should continue elucidating novel factors affecting the tumor micro-environment and promoting metastasis.”

Page 12:

“Drug resistance is also mediated by the tumor microenvironment and through transcriptomic changes. Stromal cell secretion of growth factors has been shown to promote MAPK/ERK and PI3K/AKT pathways. [119][120] Other reviews have discussed in more detail the ways the tumor microenvironment promotes resistance. [121]”

We very much look forward to your further review and comments.

Round 2

Reviewer 1 Report

Revision is acceptable.